# The Controversial Effect of Antibiotics on Methicillin-Sensitive *S. aureus*: A Comparative In Vitro Study

**DOI:** 10.3390/ijms242216308

**Published:** 2023-11-14

**Authors:** Valeria C. J. Hackemann, Stefan Hagel, Klaus D. Jandt, Jürgen Rödel, Bettina Löffler, Lorena Tuchscherr

**Affiliations:** 1Institute for Medical Microbiology, Jena University Hospital, 07747 Jena, Germany; vcj.priemer@gmail.com (V.C.J.H.); juergen.roedel@med.uni-jena.de (J.R.); bettina.loeffler@med.uni-jena.de (B.L.); 2Institute of Infectious Diseases and Infection Control, Jena University Hospital, 07747 Jena, Germany; stefan.hagel@med.uni-jena.de; 3Otto Schott Institute of Materials Research (OSIM), Friedrich Schiller University Jena, 07743 Jena, Germany; p1jakl@uni-jena.de; 4Jena School for Microbial Communication (JSMC), 07743 Jena, Germany

**Keywords:** *Staphylococcus aureus*, bacteremia, cefazolin, flucloxacillin, rifampicin, fosfomycin, small colony variants, bacterial persistence

## Abstract

Methicillin-sensitive *Staphylococcus* (*S.*) *aureus* (MSSA) bacteremia remains a global challenge, despite the availability of antibiotics. Primary treatments include β-lactam agents such as cefazolin and flucloxacillin. Ongoing discussions have focused on the potential synergistic effects of combining these agents with rifampicin or fosfomycin to combat infections associated with biofilm formation. Managing staphylococcal infections is challenging due to antibacterial resistance, biofilms, and *S. aureus*’s ability to invade and replicate within host cells. Intracellular invasion shields the bacteria from antibacterial agents and the immune system, often leading to incomplete bacterial clearance and chronic infections. Additionally, *S. aureus* can assume a dormant phenotype, known as the small colony variant (SCV), further complicating eradication and promoting persistence. This study investigated the impact of antibiotic combinations on the persistence of *S. aureus* 6850 and its stable small colony variant (SCV strain JB1) focusing on intracellular survival and biofilm formation. The results from the wild-type strain 6850 demonstrate that β-lactams combined with RIF effectively eliminated biofilms and intracellular bacteria but tend to select for SCVs in planktonic culture and host cells. Higher antibiotic concentrations were associated with an increase in the zeta potential of *S. aureus*, suggesting reduced membrane permeability to antimicrobials. When using the stable SCV mutant strain JB1, antibiotic combinations with rifampicin successfully cleared planktonic bacteria and biofilms but failed to eradicate intracellular bacteria. Given these findings, it is reasonable to report that β-lactams combined with rifampicin represent the optimal treatment for MSSA bacteremia. However, caution is warranted when employing this treatment over an extended period, as it may elevate the risk of selecting for small colony variants (SCVs) and, consequently, promoting bacterial persistence.

## 1. Introduction

*Staphylococcus aureus* is a significant causative agent in severe cases of both community-acquired and healthcare-associated bacteremia [1]. The incidence of *S. aureus* bacteremia is on the rise, with reported rates ranging from 10 to 30 cases per 100,000 person-years. This infection is of great concern due to its high mortality rate, which varies from 20% to 33% within 90 days of diagnosis. Several factors contribute to this increased mortality, including advanced age and a higher prevalence of underlying health conditions [2].

Despite the use of antimicrobial drugs, there remains a substantial mortality risk associated with *S. aureus* bacteremia [3]. Inappropriate treatment can lead to the recurrence of *S. aureus* bacteremia and the development of serious complications. One key factor contributing to inappropriate antimicrobial treatment is the uncertainty about the optimal treatment duration, which depends on the presence of metastatic infections during *S. aureus* bacteremia [4]. Patients with community-acquired *S. aureus* bacteremia and those with prolonged bacteremia are at an elevated risk of developing secondary infections. These secondary infections include conditions such as infective endocarditis, vertebral osteomyelitis, iliopsoas abscess, and septic arthritis [5].

Despite the long history of antibiotic development, clinicians still face challenges in effectively treating bacteremia (SAB), with a notable number of treatment failures.

The choice of antibiotics for the treatment of bloodstream infections depends on whether *S. aureus* is sensitive to oxacillin (MSSA) or resistant (MRSA). In cases of bloodstream infections induced by methicillin (oxacillin)-sensitive *S. aureus* (MSSA), β-lactam antibiotics with high activity against *S.aureus* are the preferred antimicrobial agents. These antibiotics, such as penicillins and cephalosporins, are covalent inhibitors that focus on bacterial-penicillin-binding proteins, interfering with peptidoglycan synthesis [6]. The best outcomes are achieved with anti-staphylococcal penicillins, such as flucloxacillin, and first-generation cephalosporins such as cefazolin [7].

Recently, the relevance of penicillin allergies has been discussed. In cases of immediate-type (IgE-mediated) penicillin allergies, daptomycin is recommended as an alternative to β-lactam antibiotics. Vancomycin use has been associated with increased mortality compared with β-lactam antibiotics; thus, it not recommended for the definitive treatment of MSSA bloodstream infections [4,7].

Rifampicin inhibits the enzyme responsible for DNA transcription, whereas fosfomycin interferes with bacterial cell wall synthesis [8,9,10]. The role of combination therapy, particularly β-lactam antibiotics in addition to rifampicin or fosfomycin for treating MSSA in the context of staphylococcal bloodstream infections, is controversial. Combination therapy could enhance bactericidal activity and lead to synergistic effects. However, clinical studies have not consistently demonstrated the benefits of routine combination therapy for all patients with staphylococcal bloodstream infections [8,11,12]. Nonetheless, some patients with staphylococcal bloodstream infections experience recurrent infections over time, potentially due to persistent microorganism reservoirs that were not eliminated by conventional treatment with β-lactamase antibiotics [8]. Rifampicin and fosfomycin have excellent penetration into host cells for the elimination of intracellular bacteria and biofilms [13,14]. Hence, employing combination therapies involving these antimicrobials could effectively eliminate *S. aureus* reservoirs, consequently diminishing the risk of recurrent infections.

*S. aureus*, a highly adaptable pathogen, deploys a diverse array of virulence mechanisms and persistence strategies, which significantly enhance its capacity for successful colonization and infection *S. aureus* can infect and reproduce within host cells [15], has the ability to change to a quiescent phenotype known as the small colony variant (SCV) [16], or form extracellular biofilms [17]. These characteristics collectively endow *S. aureus* with the capacity to maintain persistent infections, even in the face of robust host immune responses [18]. Traditionally considered an extracellular pathogen, recent findings have revealed the ability of *S. aureus* to evade detection while residing within a diverse spectrum of host cells. These include professional and non-professional phagocytic cells such as endothelial and epithelial cells. By residing inside these cells, the pathogen gains protection from the effects of multiple antibiotics, especially β-lactam antibiotics, which have limited penetration capacity into cellular environments [19].

Biofilms are complex and structured communities of microorganisms embedded in a self-produced matrix of extracellular polymeric substances that provide a mechanism by which bacteria can fully evade the host immune response and the effects of antibiotic treatment [20]. The remarkable ability of biofilms to tolerate even high-dose antibiotics, termed recalcitrance, results in persistent infections [17]. Furthermore, the dispersion of biofilms may facilitate the hematogenous spread of bacteria to other tissues, leading to metastatic infections. As a result, eradicating bacterial biofilms poses a continuing challenge for clinicians [21].

*S. aureus* displays a remarkable capability to transition into various cell types during chronic infections, including small colony variants (SCVs), persisters, L-forms, and biofilms. Each of these cell types exhibits reduced fitness but diverges in the specific molecular or genetic pathways governing their development [22]. SCVs in particular represent a subpopulation of bacteria characterized by stress resistance, slow growth, reduced hemolytic activity, and lack of pigmentation [22,23]. This phenotypic switch in *S. aureus* occurs under specific stress conditions induced by factors such as antibiotic treatment, low temperature, pH variations, osmotic changes, or an acidic intracellular environment [24,25,26]. SCVs exhibit lower immunogenicity and induce reduced cytotoxicity due to their diminished toxin production, which enables them to persist intracellularly [16,27]. This persistence and reduced immunogenicity contribute significantly to the higher incidence of *S. aureus* persistence or recurrence, setting it apart from other bacteria such as streptococci, where SCVs are seldom encountered [28,29].

The zeta potential is a critical electrochemical parameter denoting the electrical potential difference at the hydrodynamic boundary between the surrounding aqueous medium and the stationary layer of fluid adhering to the bacterial cell surface. This parameter plays a pivotal role in the regulation of cellular function and provides valuable insights into the structural and electrical characteristics of the cell surface. Additionally, it serves as an indicator of the stability and potential damage to bacterial membranes [30]. *S. aureus*, being negatively charged, has a highly negative zeta potential due to anionic phosphate groups in the glycerol phosphate repeating units of teichoic acids [31,32]. It is hypothesized that SCVs have an altered cell wall thickness, probably due to impaired peptidoglycan synthesis, resulting in a different (less negative) surface charge [33]. Moreover, the zeta potential serves as a valuable tool for measuring the effects of antibiotics on bacterial membranes [34,35].

This study was designed to investigate the effects of antibiotics and their combinations on planktonic bacterial growth, biofilm formation, intracellular bacteria and SCVs, which represent possible bacterial reservoirs in bloodstream infections. To imitate acute and chronic infection stages, we utilized the laboratory wild-type strain 6850, and a stable SCV mutant strain JB1 derived from 6850. By employing these laboratory models, we quantified the effects of cefazolin and flucloxacillin treatments, as well as the antibiotic combinations with either fosfomycin or rifampicin, the most frequent regimens for treating MSSA bacteremia [7,36]. Our objective was to identify the most efficient treatment options, thereby guiding the most effective antibiotics for combating *S. aureus* bacteremia.

## 2. Results

### 2.1. S. aureus Strains 6850 and JB1 Were Susceptibile to All Antibiotic Combinations during Planktonic Growth

To assess the direct impact of antibiotics and their combinations in vitro, we conducted experiments on planktonic bacteria. The sensitivity of the strains to the selected antibiotics was determined by calculating the MIC for each strain. Both the *S. aureus* wild-type strain 6850 and its SCV counterpart, JB1, which is a stable menadione auxotrophic SCV variant, displayed susceptibility to all tested antibiotics (Table 1A). Additionally, synergy assays were conducted for each antibiotic combination. Fractional inhibitory concentration indices (FICIs) revealed synergy between flucloxacillin and rifampicin for *S. aureus* 6850 and cefazolin and rifampicin for its SCV derivative JB1 (Table 1B).

### 2.2. Antibiotic Combinations Containing Rifampicin Effectively Killed Intracellular Bacteria but Not the Intracellular SCV S. aureus JB1

*S. aureus* has the ability to invade and replicate within host cells, where it gains protection from the antimicrobial effects of many antibiotics due to limited intracellular penetration. This phenomenon occurs during the chronic phase of infection and contributes to therapeutic failure. To investigate the antimicrobial efficacy of cefazolin and flucloxacillin, either alone or in combination with rifampicin or fosfomycin, endothelial cells (Ea.hy926; ATCC CRL-2922) were infected with *S. aureus* strains 6850 and JB1. One day after infection, the infected cells were treated with cell medium containing varying concentrations of antibiotics alone or in combination. Notably, antibiotic combinations containing rifampicin significantly reduced intracellular *S. aureus* 6850 compared with untreated cells at all MOIs tested (Figure 1A). Only at high MOIs (5, 10 and 20) was a significant reduction also observed in combinations with fosfomycin.

However, intracellular colony-forming units (CFU) of JB1 were not reduced by any of the tested treatments (Figure 1B). These results suggest a synergistic effect of rifampicin at all MICs tested and with fosfomycin at high MICs on wild-type intracellular *S. aureus*, but not on SCVs.

### 2.3. Antibiotic Combinations with Rifampicin Promote the Selection of SCVs

SCVs represent a bacterial phenotype that exhibits reduced susceptibility to various antimicrobials due to their low metabolic activity and limited membrane permeability [37]. Upon analyzing the proportion of SCVs relative to normal colony variants for the wild-type *S. aureus* strain 6850, combinations of fluxcloxacillin with rifampicin significantly increased the selection of SCVs, both in planktonic culture (Figure 2A) and intracellularly (Figure 2B). Furthermore, a consistent occurrence of elevated SCVs was observed only in planktonic culture with high concentrations (MICs 10 and 20) of antibiotics combined with rifampicin or fosfomycin (Figure 2A).

Assessment of auxotrophy for specific compounds, including menadione, hemin, CO_2_ and thymidine, was performed on several SCVs recovered from planktonic and intracellular experiments. However, non-specific auxotrophy was observed across all observed SCVs and experimental conditions (Appendix A).

### 2.4. Antibiotics Successfully Halt Biofilm Formation in S. aureus WT and SCV 

The prevention of biofilm formation was tested by adding various concentrations of antibiotics directly to bacterial cultures before biofilm growth. All antibiotics were effective in preventing biofilm formation for both strains, particularly at high doses, compared with the untreated control (Figure 3: *S. aureus* 6850 (A) and JB1 (B)).

### 2.5. Successful Biofilm Eradication via Rifampicin and Fosfomycin Antibiotic Combinations, JB1 Biofilms Respond Solely to Rifampicin

Biofilms were cultivated using both strains, and after 48 h of growth, they were subjected to monotherapy and combination antibiotic treatments. Biofilm eradication on *S. aureus* 6850 by antibiotics was most successful with combinations containing rifampicin and cefazolin with fosfomycin (Figure 4A). However, JB1 biofilms were barely affected by any treatment except for high doses containing rifampicin (Figure 4B). Furthermore, increased biofilm formation was observed when biofilms from both strains were exposed to flucloxacillin alone (Figure 4A,B) or, in the case of *S. aureus* 6850, flucloxacillin plus fosfomycin (Figure 4A).

### 2.6. The Zeta Potential of All Strains Was Less Negative When Preincubated with Antibiotics

The zeta potential of *S. aureus* 6850 was significantly more negative than the zeta potential of the SCV strain JB1 (Figure 5A). When pre-incubated with antibiotics, the zeta potential of the *S. aureus* wild-type strain 6850 became significantly less negative than that of the untreated control (Figure 5B), suggesting a modification in the permeability of its membrane.

## 3. Discussion

*S. aureus* infections can persist despite adherence to treatment guidelines. A combination of β-lactamase antibiotics with fosfomycin or rifampicin is under discussion for the treatment of MSSA bloodstream infections and possible metastatic associated infections such as infective endocarditis, vertebral osteomyelitis, iliopsoas abscess and septic arthritis. Several trials have reported better patient outcomes with combination therapy. However, other studies have shown that combination therapy does not improve the treatment of MSSA bacteremia or its possible risk of late secondary infection [4,7,8,11,12]. The ability of *S. aureus* to switch to SCVs, form biofilms and/or hide intracellularly may interfere with the efficacy of antimicrobial therapy and may be related to the discrepancies between these studies [22,23,38,39]. Our in vitro study has provided valuable insights. Antibiotics can inadvertently contribute to the development of chronic infections [40]. While these agents reduced the overall bacterial colony count, certain antibiotics, particularly combinations containing rifampicin, notably favored the selection of SCVs in planktonic culture—a phenotype linked to chronic infections.

The determination of the MIC is the gold standard for assessing the antimicrobial susceptibility of pathogenic bacteria [41]. When the MIC indicates the effectiveness of a drug in inhibiting the growth of a specific organism, it is used for the treatment of infection. However, in vitro susceptibility does not guarantee the same effect in vivo [42]. In our study, both *S. aureus* 6850 and JB1 showed susceptibility to all antibiotics tested. However, different results were observed when these antibiotics were tested against biofilm or intracellular bacteria.

*S. aureus* possesses the ability to hide itself intracellularly, evading antimicrobial treatment and the immune system, especially in challenging infections such as osteomyelitis [40]. Our study shows that only combinations with rifampicin effectively reduced the intracellular bacterial burden. Combinations with fosfomycin were effective in reducing intracellular bacterial load, but mainly at higher doses. In particular, the combination of flucloxacillin with rifampicin was found to promote the formation of intracellular SCVs. When host cells were exposed to the stable mutant SCV JB1, none of the antibiotics substantially eradicated infection, including combinations with rifampicin, a drug known for its intracellular bacterial eradication potential [43]. This highlights the robust protection of SCVs against antibiotics within cells, as previously documented [27]. In addition, the induction of SCVs by antibiotics such as rifampicin further promotes intracellular bacterial persistence and the formation of bacterial reservoirs. This may contribute to the recurrence of infection observed in some patients after antimicrobial treatment.

When *S. aureus* 6850 biofilms were treated with antibiotic combinations with rifampicin or with cefazolin combined with fosfomycin, a significant reduction in biofilms was observed. None of the combination treatments showed a significant reduction in JB1 biofilms. In contrast, flucloxacillin monotherapy increased JB1 biofilm formation. Considering that biofilms play a critical role in most bacterial infections and are a major contributor to treatment failure [44], the capacity of SCVs to generate highly resilient biofilms should be a key consideration in antibiotic selection decisions. 

Pretreating the strains with high concentrations of antibiotics led to a less negative zeta potential of all strains, signaling a shift in cell surface charge, thereby increasing aggregation as described before in biofilms [45]. This phenomenon can serve as an indicator of diminished membrane permeability in SCVs toward antibacterial compounds and offers an explanation for the reduced susceptibility of *S. aureus* to antibiotics, especially in chronic infections [35]. Our observation also revealed that JB1 exhibited a less negative zeta potential compared to wild-type strains elucidating why eradicating JB1 proved exceptionally challenging compared to wild-type strains. 

However, this study has limitations. All experiments were confined to in vitro settings, and the effects of antimicrobials may differ in vivo due to the influence of the host immune response, which could introduce other changes in the interplay between bacteria and antibiotics not considered in our findings. Further investigations using animal models or professional phagocytes are essential to deepen our understanding of antibiotic interactions with the immune system and their effectiveness against *S. aureus*. Recent research has also indicated that antibiotics can dampen adaptive immunity, potentially fostering recurrent infections [46,47,48].

Additionally, due to the unstable phenotype of clinical SCV strains recovered from our assays, we were unable to conduct experiments with them. As a substitute, the stable mutant SCV-strain JB1 was utilized throughout all the experiments. Nevertheless, prior studies have demonstrated that clinical SCVs and stable mutant SCV strains exhibited similarities in terms of growth and metabolic state, irrespective of their auxotrophisms [49].

Taken together, these findings shed light on the delicate balance between eradicating bacteria with antibiotics and unintentionally selecting persistent cell variants through antibiotic exposure.

## 4. Material and Methods

**Bacterial strains:** For all experiments, the highly virulent *S. aureus* strain 6850 and its SCV derivative JB1 were used [27]. Broth dilution assays were performed using a sterile 96-well plate with twofold dilutions of antibiotics alone and combinations of cefazolin and rifampicin, cefazolin and fosfomycin, flucloxacillin and rifampicin, or flucloxacillin and fosfomycin, set up in a checkerboard pattern. Bacterial cultures were inoculated in the wells at a concentration of 5 × 10^5^ CFU/mL in Mueller-Hinton broth (Oxoid, Waltham, MA, USA). After incubation of wild-type strains for 18 h and JB1 for 36 h at 37 °C with shaking, absorbance was measured at 578 nm using a microplate reader (Infinite^®^ 200 PRO, Tecan, Männedorf, Switzerland). The well with the lowest antibiotic concentration showing no turbidity was determined as the minimal inhibitory concentration (MIC), and the well with the highest antibiotic concentration exhibiting visible growth was identified as the sub-MIC. The fractional inhibitory concentration index (FICI) was calculated to investigate whether both antibiotics are synergistic, indifferent, or antagonistic, as described elsewhere [50]. The exact formula used was as follows: A/MIC_A_ + B/MIC_B_ = FIC_A_ + FIC_B_ = FICI,(1)
where A is the concentration of antibiotic A and B is the concentration of antibiotic B. MICs were further confirmed by plating samples and confirming bacterial growth reduction by at least 99%. Each antibiotic combination was repeated six times per strain and antibiotic combination, as previously recommended [51]. After measuring the growth dynamics of the wild-type strain, the inoculum was plated using serial dilutions on Columbia blood agar (10^−1^ to 10^−8^) [51]. Detection of SCVs on plates was performed by observing the plates for up to 72 h. The number of SCVs was calculated in relation to the number of wild-type colonies (SCV/Wt).

**Cell culture:** The endothelial cell line Ea.hy926 (ATCC CRL-2922) was employed for cell experiments [52]. These Ea.hy926 cells, a permanent human cell line resulting from the fusion of A549 cells with umbilical vein endothelial cells, express factor VIII-related antigen, making them suitable for investigating nonprofessional phagocytes or the endothelium. This cell line was chosen because it represents the first nonprofessional phagocytes in contact with *S. aureus*.

To test the effect of cefazolin (Hikma, London, UK), Flucloxacillin (Stragen, Cologne, Germany), and their combination with fosfomycin (infectoPharm, Heppenheim, Germany) and rifampicin (Riemser, Berlin, Germany) on bacteria within host cells, infected endothelial cells were treated with different concentrations of the antibiotics, according to their MIC detected via the checkerboard assay and CLSI broth dilution method. Initially, all strains were cultivated overnight in Mueller Hinton medium, OD = 1 was adjusted, and the CFU/mL was calculated by plating the obtained OD = 1 on blood agar plates. Cells were grown to 80% confluence in six-well plates, and the multiplicity of infection (MOI) was calculated. Cells were infected using an invasion medium consisting of DMEM (PAN biotech, Aidenbach, Germany) with 10% fetal bovine serum (FBS, Bio & Sell, Feucht, Germany), 1% HSA (Octapharma, Lachen, Switzerland), and 10 mM HEPES (Sigma Aldrich, St. Louis, MI, USA) with at an MOI of 100 and then incubated for 90 min at 37 °C with 5% CO_2_. Then, they were washed with PBS and treated with 1 mL of a stopping medium containing DMEM with 10% FBS and 200 µg/mL lysostaphin (WAK Chemie, Steinbach i.Ts., Germany) for 30 min to kill extracellular bacteria. After that, the cells were washed, treated with a full medium containing 10% FBS and 1% penicillin/streptomycin, and incubated for 24 h at 37 °C with 5% CO_2_. The next day, cells were washed and treated with a DMEM (PAN Biotech, Aidenbach, Germany) containing 1% FBS, 1% HSA, and different concentrations of the antibiotics and their combinations according to the MIC (1×, 2×, 5× MIC, 10× MIC, and 20× MIC). Moreover, a low concentration of FBS was chosen to avoid protein binding by the antibiotics of choice. After one more day, the supernatants were plated on blood agar plates to see if the antibiotics killed bacteria that were released overnight. Then, the cells were washed with PBS (Gibco, Carlsbad, CA, USA), treated with stopping medium containing 200 µg/mL lysostaphin for 30 min, washed again with PBS, and lysed using 1 mL of icecold water. After 10 min of incubation at room temperature, the cells were scraped, centrifuged for 10 min at 4 °C at 5000 rpm, and resuspended in 1 mL PBS. Then, recovered intracellular bacterial solutions were plated in dilution series (10^−1^ to 10^−3^) on Columbia blood agar and incubated at 37 °C. From the following day on, the plates were observed for three days to count colonies with wild-type and SCV phenotypes. For further statistical evaluation, counted SCVs were examined in a ratio with counted wild-type colonies (SCV/Wt).

**Identification of SCVs:** The SCVs were identified on solid media as SCV colonial morphotype colonies when the agar plates were incubated for 24, 48 and 72 h. They were identified as a slow-growing population (1/10 the size of wild-type colonies), lacking pigmentation and hemolysis. To confirm that they were *S. aureus*, we performed PCR with the following primers to amplify the *nuc* genes fw “gcg att gat ggt gat acg gtt” and rev “agc caa gcc ttg acg aac taa agc” [53].

**Auxotrophism test:** To find possible auxotrophisms, disc diffusion assays were performed as described elsewhere [54]. Control strains included the menadione auxotroph SCV mutant JB1, the hemB mutant from 6850 *S. aureus* strain IIb13 [27], and the thymidine mutant SCV strain BF2418.

**Biofilm experiments:** Biofilm assays were conducted to examine the impact of β-lactams on biofilm formation and eradication. Combinations of antibiotics with rifampicin or fosfomycin were also tested for potential synergistic effects. Biofilm mass was measured using the crystal violet staining method [55], and the results were compared with untreated samples. The controls were set equal to 100%. The exact equation was as follows:Biofilm (%) = OD_(treated sample)_ × 100/OD_(untreated sample)_.(2)

The biofilm-producing *S. epidermidis* strain RP62A served as a positive control. Experiments were repeated five times in quadruplicate.

**Biofilm prevention:** To study antibiotic prevention of biofilm formation, strains were cultured overnight in TSB (Oxoid, Waltham, MA, USA) with 0.25% glucose (Carl Roth, Karlsruhe, Germany). Then, each well was inoculated with 1 µL of the overnight culture and incubated for 48 h at 37 °C. After that, the wells were carefully washed twice with PBS and stained with 100 µL of 1% crystal violet per well. After 15 min of incubation, each well was washed thrice with PBS, 100 µL ethanol/acetone (in a ratio of 80:20) was added, the plate was incubated for 10 min, and the absorbance was measured with an Infinite^®^ 200 PRO microplate reader (Tecan, Männedorf, Switzerland) at 550 nm as an indication of the biofilm mass.

**Biofilm eradication:** To further investigate the antimicrobial effect on already formed biofilm, all strains were, as explained above, prepared in an overnight culture in 3 mL TSB with 0.25% glucose, and a sterile 96-microtiter plate with 200 µL TSB-Glucose and 1 µL of the overnight culture was prepared and incubated for 48 h. After that, biofilms were washed to remove planktonic bacteria, and antibiotics were added to each well at concentrations of 0×, 1×, 2×, 5×, 10×, and 20× the MIC according to the MIC found in the synergy assays. Then, they were incubated for another 24 h. The next day, the plate was washed twice and stained with crystal violet, and the absorbance was measured as described before.

**Zeta potential:** Zeta potential measurements were conducted to assess the influence of antibiotics on bacterial membrane potential. Fixed bacterial samples were diluted in 0.5 mL PBS with 0.5 mL Milli-Q water to obtain a stable pH of 7.2. This is important as the zeta potential changes with pH changes. The zeta potential was measured using Zetasizer Nano ZS 90 device (Malvern Panalytical, Malvern, UK) at room temperature (25 °C).

**Statistical analyses:** Two-way ANOVA, one-way ANOVA followed by Dunnett’s multiple comparison test, and an unpaired *t*-test were performed using GraphPad Prism version 10.0.0 for Windows (GraphPad Software, Boston, MA, USA).

## 5. Conclusions

Based on our in vitro results, the concurrent administration of cefazolin or flucloxacillin with rifampicin effectively eliminated *S. aureus*. Nevertheless, the emergence of SCVs was possibly enhanced, and these SCVs have the potential to trigger infection recurrences, thereby contributing to prolonged and persistent infection states.

## Figures and Tables

**Figure 1 ijms-24-16308-f001:**
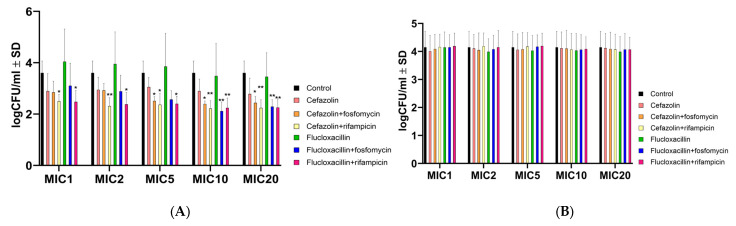
Efficacy of antibiotic monotherapy or combinations for the treatment of intracellular *S. aureus* 6850 (**A**) and JB1 (**B**), expressed as mean log CFU versus untreated control ± SD. Statistical analyses were performed using two-way ANOVA followed by Dunnett’s multiple comparison test. The significance levels obtained in the post hoc analyses are indicated by asterisks. According to the results of the analyses, the differences are either not significant (*p* > 0.05) or significant (* *p* < 0.05; ** *p* < 0.01). Results are from 5 independent experiments.

**Figure 2 ijms-24-16308-f002:**
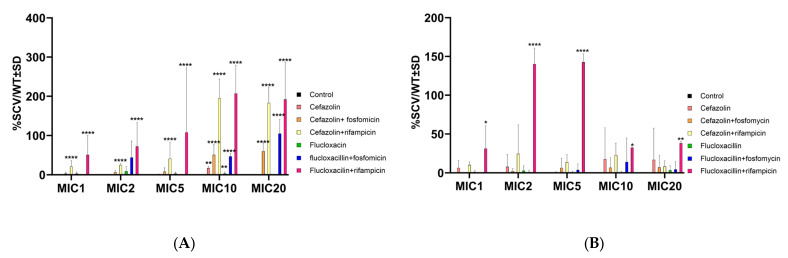
Ratio of small colony variants to wild-type colonies (SCV/WT) shown as mean ± SD of *S. aureus* 6850 in planktonic culture (**A**) and intracellular (**B**). No SCVs were found in the controls (=0). Statistical analyses were performed using two-way ANOVA followed by Dunnett’s multiple comparison test with respect to the untreated control. Significance levels of post hoc analyses are indicated by asterisks. Significance levels of post-hoc analyses are indicated by asterisks. According to the results of the analyses, the differences are either significant (*p* > 0.05) or not significant (* *p* < 0.05; ** *p* < 0.01; and **** *p* < 0.0001). The results were obtained from 5 independent experiments.

**Figure 3 ijms-24-16308-f003:**
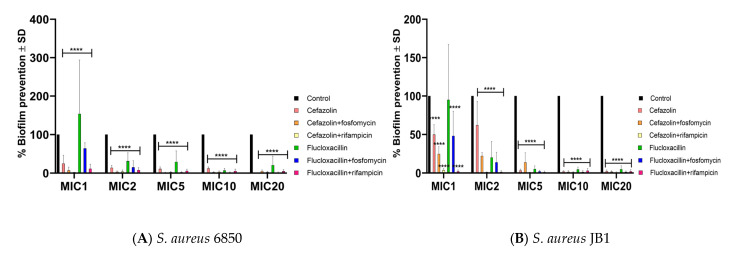
Biofilm inhibition reported as the mean for *S. aureus* 6850 (**A**) and JB1 (**B**) relative to the untreated control (OD_(treated sample)_ × 100/OD_(untreated sample)_). The untreated control was set to 100%. Statistical analyses were performed using two-way ANOVA followed by Dunnett’s multiple comparison test. The significance of the post-hoc analyses is indicated by asterisks when the biofilm mass was reduced and by hash marks when it was induced. According to the results of the analyses, the differences are either not significant (*p* > 0.05) and **** *p* < 0.0001). Results are from 5 independent experiments.

**Figure 4 ijms-24-16308-f004:**
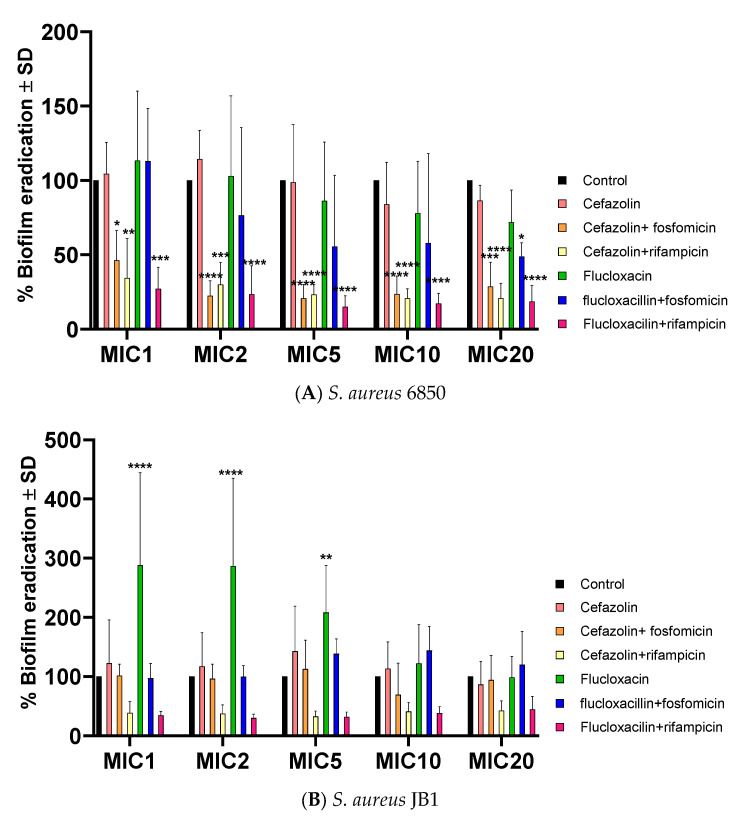
Biofilm eradication for *S. aureus* 6850 (**A**) and JB1 (**B**) expressed as the mean of the OD relative to the untreated control ± SD (OD_(treated sample)_ × 100/OD_(untreated sample)_). All values represent at least five independent experiments (*n* = 5). The untreated control was set to 100%. Statistical analyses were performed using two-way ANOVA followed by Dunnett’s multiple comparison test. Significance levels of post-hoc analyses are indicated by asterisks when biofilm mass was reduced or hash marks when it was induced. According to the results of the analyses, the differences are either not significant (*p* > 0.05) or significant (* *p* < 0.05; ** *p* < 0.01; *** *p* < 0.001 and **** *p* < 0.0001).

**Figure 5 ijms-24-16308-f005:**
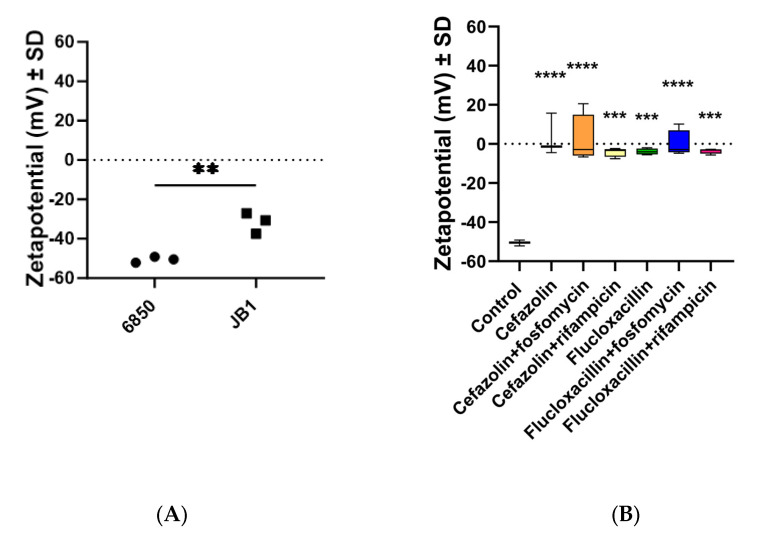
Zeta potential in mV ± SD. A: Zeta potential of *S. aureus* wild-type strain 6850 and SCV strain JB1 without antibiotic pre-treatment. The SCV strain shows significantly less negative zeta potential than the wild-type strains. *n* = 3. B: Zeta potential of 6850 at 10 × MIC. All treatments resulted in a significant increase in zeta potential. *n* = 4. Each point in the graph represents an independent experiment. Statistical analyses were performed with unpaired *t*-test (**A**) or one-way ANOVA followed by Dunnett’s multiple comparison test (**B**). According to the results of the post-hoc analyses, the significance is indicated by asterisks. (** *p* < 0.01; *** *p* < 0.001; **** *p* < 0.0001).

**Table 1 ijms-24-16308-t001:** (A) MICs reported as the mean (mg/L) ± SD for *S. aureus* 6850 and JB1. The table shows the results of six independent experiments (*n* = 6). (B) Synergy assays with MICs reported as the mean (mg/L) ± SD and fractional inhibitory concentration indices (FICI) for *S. aureus* 6850 and JB1. The first number in each case refers to the MIC of the first antibiotic named in the corresponding row. Additionally, fractional inhibitory concentration indices (FICIs) are shown for all antibiotic combinations and 6850 and JB1. An index ≤ 0.5 shows synergy, whereas an index > 0.5 and ≤4.0 indicates the indifference of the two components. An index of ≥4.0 indicates antagonism.

(**A**)				
**Antibiotics**	**6850 MIC**		JB1 MIC	
**Cefazolin**	**0.5**		0.08 ± 0.05	
**Flucloxacillin**	**0.17 ± 0.09**		0.09 ± 0.04	
**Fosfomycin**	**0.85 ± 0.69**		0.4 ± 0.2	
**Rifampicin**	**0.015**		0.015	
(**B**)				
**Antibiotic Combinations**	6850 MIC	6850 FICI	JB1 MIC	JB1 FICI
**Cefazolin + fosfomycin**	0.5 ± 0.01/ 1 ± 0.26	indifferent	0.16 ± 0.1/ 0.12 ± 0.11	indifferent
**Cefazolin + rifampicin**	0.08 ± 0.08/ 0.03 ± 0.02	indifferent	0.1 ± 0.5/ 0.002 ± 0.001	synergistic
**Flucloxacillin + fosfomycin**	0.375 ± 0.323/ 1 ± 1.3	indifferent	0.17 ± 0.1/ 0.35 ± 0.36	indifferent
**Flucloxacillin + rifampicin**	0.05 ± 0.02/ 0.015 ± 0.008	synergistic	0.15 ± 0.09/ 0.008 ± 0.006	indifferent

## Data Availability

Data are contained within the article and Appendix A.

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
