# Peer review of "The Controversial Effect of Antibiotics on Methicillin-Sensitive S. aureus: A Comparative In Vitro Study"

_ijms, 2023, doi:10.3390/ijms242216308_

Round 1

Reviewer 1 Report

Comments and Suggestions for Authors

The experimental study by Hackemann and coauthors is devoted to deciphering biological mechanisms underlying Staphylococcus aureus infections and combination antimicrobial therapy. A paper of this kind is expected to inform clinical studies and therefore is very important. Unfortunately, the manuscript has a number of major flaws. The experimental part of the paper deals with biofilm-related infections (e.g. Section 2.4), but the therapeutic compounds selected for the study are relevant for bloodstream infections (Lines 74-79). Please see papers [1] and [2] for the overview of biofilm-related S. aureus infections. They are typically chronic and often associated with indwelling medical devices. Re-think your results and prepare new Introduction and Discussion having in mind this difference. Find recent clinical literature and make clear points regarding what should be changed in practice. My another major concern relates to poor efficacy of antibiotic combinations in vivo [3, 4]. Clinical results of this kind must be mentioned in the Discussion. Does the study prove the opposite? The insufficient conceptualization of the study design prevented me from assessing the scientific soundness of the MS.

I strongly recommend to ask an experienced colleague to check the text for scientific accuracy.

The following statements from the Introduction can be considered inappropriate:

Line 37. “a resilient pathogen” means “an elastic pathogen” or “a buoyant pathogen”.

Lines 40-41. “These attributes collectively enable S. aureus to sustain persistent infections”. I don’t think S. aureus tolerates or supports infections. Rather, it causes infections.

Line 44. “non-professional cell types”. Are there professional cell types?

Line 45. “Biofilms, three-dimensional extracellular bacterial aggregates”. Any possible definition of biofilms will include the extracellular matrix.

This can be found just in the first two paragraphs. My overall impression is that the text cannot be improved sufficiently within the frames of peer review process. When there are mistakes on nearly each line, reviewers cannot spot all of them.

Minor points

The title should be amended to reflect the principal finding(s) of the study more precisely. Current title made me think there were a number of methicillin-sensitive bacterial strains eradicated by a number of other antibiotics.

In Abstract, check if all abbreviations are necessary. For example, cefazolin and fosfomycin are mentioned only once. Line 11. Inconsistent use of italics. Line 22, is it necessary to mention the exact strain number here?

Lines 31, 32. Remove numbers.

In the Introduction, explain why the study is focused on the search of efficient treatment options for susceptible bacteria. It may be due to adverse effects of methicillin and limited numbers of available administration routes for the drug. Maybe, the idea is in choosing antibiotic regimen depending on the methicillin susceptibility or its significance for classification of clinical conditions or whatever.

In the whole text, please consider writing antibiotic names in full. Currently it is difficult to read due to excessive number of abbreviations.

In the footnotes, replace the yen symbol with a more common one, like dagger or the section sign.

Present the tables as barplots.

Explain the concept of zeta potential in more details.

Check uniform spelling of words, e.g. bacteremia or bacteraemia.

Make table captions readable.

Figure 1. Fix misplaced axis titles.

[1] Moormeier DE, Bayles KW. Staphylococcus aureus biofilm: a complex developmental organism. Mol Microbiol. 2017 May;104(3):365-376. doi: 10.1111/mmi.13634. Epub 2017 Mar 8. PMID: 28142193; PMCID: PMC5397344.

[2] Suresh MK, Biswas R, Biswas L. An update on recent developments in the prevention and treatment of Staphylococcus aureus biofilms. Int J Med Microbiol. 2019 Jan;309(1):1-12. doi: 10.1016/j.ijmm.2018.11.002. Epub 2018 Nov 27. PMID: 30503373.

[3] Safdar N, Handelsman J, Maki DG. Does combination antimicrobial therapy reduce mortality in Gram-negative bacteraemia? A meta-analysis. Lancet Infect Dis. 2004 Aug;4(8):519-27. doi: 10.1016/S1473-3099(04)01108-9. PMID: 15288826.

[4] Tabah A, Laupland KB. Update on Staphylococcus aureus bacteraemia. Curr Opin Crit Care. 2022 Oct 1;28(5):495-504. doi: 10.1097/MCC.0000000000000974. Epub 2022 Aug 4. PMID: 35942696.

Comments on the Quality of English Language

The clarity and quality of the manuscript should be improved either by native speaker or by language editing service.

Reviewer 2 Report

Comments and Suggestions for Authors

This study describes the effect of mono- and combination chemotherapy on the inhibition of planktonic, sessile, and SCVs. Overall, it is a little lack of rationale. This needs to be described in more detail in the Introduction part. And, the results are not much explained in the result section.

[Comments]

1.     Provide MIC values.

2.     Is there any reasons for using the term “methicillin-sensitive SA”, rather than using just SA.

3.     Tables should include the numbers of bacteria in CFU, not percentages.

4.     Provide criteria on the selection of antibiotics used in this study, and furthermore which mechanisms are expected in the combination therapy

5.     SCV needs to be described in more detailed by comparing with other persister cells.

6.     SCV variants need to be confirmed by additional assays.

7.     And, proof-reading is required, for instance, - delete “has the” in Line 39.

Comments on the Quality of English Language

Moderate editing of English language required

Author Response

One correction, the MIC tables have been moved to the main text.

Round 2

Reviewer 1 Report

Comments and Suggestions for Authors

The Authors either implemented the requested changes or provided reasonable objections. I have no further comments.

Reviewer 2 Report

Comments and Suggestions for Authors

It has been well revised.

Comments on the Quality of English Language

 Minor editing of English language required